# Lycopene inhibits ER stress and apoptosis while modulating PI3K/AKT and enhancing antioxidant and anti-apoptotic proteins

**Lijun Zheng**[1☯]**, Hui Zhang**[2☯]**, Yuchao Sun**[iD][3]*

**1** Nursing Department, The Fourth Affiliated Hospital of School of Medicine, Zhejiang University, Yiwu, China, **2** School of Pharmacy, Faculty of Medicine, Macau University of Science and Technology, Macau (or Macau SAR), China, **3** The Core Facilities, The Fourth Affiliated Hospital of School of Medicine, and International School of Medicine, International Institutes of Medicine, Zhejiang University, Yiwu, China

☯ These authors contributed equally to this work.

\* 8018136@zju.edu.cn

## Abstract

Acute kidney injury (AKI) is a critical clinical syndrome with limited therapeutic options. This study investigated the renoprotective effects of lycopene, a potent antioxidant, in both in vivo and in vitro AKI models. In a murine cecal ligation and puncture (CLP)-induced sepsis-AKI model, pretreatment with lycopene (10, 20, 40 mg/kg) dose-dependently ameliorated renal histopathological damage (HE staining) and restored serum biomarkers (AST, ALT, BUN, CREA). Mechanistically, lycopene suppressed oxidative stress and apoptosis by downregulating the PI3K/Akt axis: it significantly reversed the CLP-induced upregulating of p-PI3K and p-Akt, and the pro-apoptotic proteins (Bax, Cleaved Caspase-3), while increasing Nrf 2 and SOD1. Consistent results were observed in LPS- and $H_2O_2$-induced cellular AKI models, where lycopene attenuated cell death and restored redox homeostasis in a dose-dependent manner. Immunofluorescence assays further validated these trends. Crucially, PI3K siRNA or cDNA transfection experiments confirmed that lycopene's antioxidant and anti-apoptotic effects were PI3K-dependent. Our findings highlight lycopene as a promising therapeutic agent for AKI, acting via PI3K/Akt-mediated activation of Nrf 2 to counteract oxidative damage and apoptosis. This study provides novel insights into the molecular mechanisms underlying lycopene's renoprotection and supports its potential clinical translation for sepsis or oxidative stress-associated AKI.

## Introduction

Sepsis, a dysregulated host response to infection leading to life-threatening organ dysfunction, imposes a substantial global health burden with an estimated 48.9 million cases and 11 million sepsis-related deaths annually worldwide. The kidneys

**Data availability statement:** All relevant data are within the manuscript and its Supporting information files.

**Funding:** Supported by Foundation of Zhejiang Provincial Education Department, Y202353817.

**Competing interests:** The authors have declared that no competing interests exist.

rank among the most vulnerable organs in this systemic crisis, with sepsis-associated acute kidney injury (SA-AKI) occurring in approximately 40–60% of critically ill septic patients. The development of SA-AKI dramatically increases mortality risk, prolongs ICU stay, and heightens the likelihood of progressing to chronic kidney disease, underscoring an urgent, unmet clinical need for effective therapeutic strategies. Current management remains largely supportive, relying on infection control, hemodynamic stabilization, and renal replacement therapy, while lacking specific interventions to halt the underlying molecular pathogenesis within renal cells. Against this clinical backdrop, a detailed dissection of the cellular mechanisms driving renal injury is paramount [1–5].

Sepsis-associated acute kidney injury (SA-AKI) is a lethal complication, as evidenced by a prospective study reporting a 28-day mortality rate of 32.7% in septic patients with AKI [6]. Despite this grave clinical burden, its underlying molecular mechanisms remain incompletely understood [7]. The cecal ligation and puncture (CLP)-induced murine model, a well-established preclinical paradigm for polymicrobial sepsis, reliably recapitulates the pathophysiological features of human SA-AKI, including systemic inflammation, oxidative stress, and tubular apoptosis [8–10]. Recent investigations using this model have revealed aberrant activation of the phosphatidylinositol 3-kinase (PI3K)/protein kinase B (AKT) signaling pathway, characterized by increased phosphorylation of PI3K (p-PI3K) and AKT (p-AKT), concomitant with diminished expression of nuclear factor erythroid 2-related factor 2 (Nrf 2), a master regulator of antioxidant responses [11,12]. These molecular alterations collectively exacerbate renal apoptosis and oxidative damage, highlighting the therapeutic potential of targeting these pathways.

The PI3K/AKT axis plays a paradoxical role in cellular homeostasis, functioning as a "double-edged sword" depending on contextual activation dynamics and disease stages [13–15]. Under physiological conditions, PI3K catalyzes the conversion of phosphatidylinositol-4,5-bisphosphate (PIP2) to phosphatidylinositol-3,4,5-trisphosphate (PIP3), recruiting AKT to the cell membrane for activation [16]. Activated AKT promotes cell survival by inhibiting pro-apoptotic proteins (e.g., BAD, caspase-9) and enhancing nutrient uptake [17]. However, hyperactivation of this pathway under pathological stress, such as sepsis, may paradoxically drive apoptosis through feedback mechanisms [18]. Excessive PI3K/AKT signaling can induce mitochondrial dysfunction, reactive oxygen species (ROS) overproduction, and endoplasmic reticulum stress, ultimately triggering caspase-3-mediated apoptosis [19,20]. Furthermore, PI3K isoforms (e.g., class IA p110α vs. class III Vps34) exhibit divergent roles: while class IA PI3K supports survival, class III variants may promote autophagy-lysosomal dysregulation in sepsis [21,22]. In the CLP model, sustained PI3K/AKT activation likely reflects a maladaptive response to unresolved inflammation, creating a vicious cycle of oxidative damage and tubular cell death.

Compounding this imbalance, the observed suppression of Nrf 2 in CLP-induced nephropathy impairs the transcription of antioxidant enzymes (e.g., HO-1, NQO1), leaving renal cells vulnerable to ROS accumulation [23,24]. Lycopene (Lyc), a potent lipophilic carotenoid abundant in tomatoes, emerges as a promising therapeutic

candidate due to its multimodal pharmacodynamic profile [25]. Beyond direct free radical scavenging, lycopene modulates key signaling pathways by: 1) Inhibiting NF-κB-driven inflammation through interference with IκB kinase activity; 2) Enhancing Nrf 2 nuclear translocation via PKC-δ and MAPK pathway regulation, thereby restoring antioxidant defenses; and 3) Paradoxically normalizing PI3K/AKT hyperactivity by suppressing upstream activators (e.g., growth factor receptors) while preserving baseline survival signals [26–28]. This nuanced modulation may disrupt the pathological PI3K/AKT-Nrf-2 axis crosstalk without inducing widespread pathway inhibition.

This study investigates the hypothesis that lycopene ameliorates CLP-induced renal injury by rectifying the imbalance between pro-survival and pro-apoptotic PI3K/AKT signaling while augmenting Nrf-2-mediated antioxidant responses. By elucidating the compound's dual regulatory effects on this critical "molecular switch," the findings may advance targeted therapeutic strategies for SA-AKI. To achieve these aims, our experimental strategy was threefold: First, we employed in vivo models to assess the overall renal histopathological changes (by H&E staining) and functional outcomes. Second, we utilized a combination of molecular techniques, including Western blotting and immunohistochemical and immunofluorescence staining, to delineate the activation states of key signaling pathways (e.g., PI3K/AKT/Nrf-2) and the expression of related proteins within renal tissues. Finally, in cellular models, we confirmed the protective effect of lycopene on cell viability (by CCK-8 assay) and investigated its role in mitigating apoptosis through flow cytometry and JC-1 assay, with the specific molecular mechanisms further verified by gene silencing techniques.

## Materials and methods

### Establishment of sepsis model and experimental grouping

Forty specific pathogen-free (SPF) female BALB/c mice (aged 8–10 weeks, body weight 20–25 g) were purchased from the Laboratory Animal Center of Zhejiang University (Animal Use License No. SYXK (Zhe) 2018−0005). The animal study protocol was approved by the Institutional Animal Care and Use Committee of Zhejiang University (Ethics Approval No. ZJU20250874). All mice were housed under standardized laboratory conditions (temperature $22 \pm 2°C$, humidity $50 \pm 10\%$, 12 h light/dark cycle) with ad libitum access to food and water throughout the experimental period.

Following a 12-hour preoperative fast (with free access to water), the cecal ligation and puncture (CLP) procedure was performed to induce polymicrobial sepsis in mice. Under isoflurane anesthesia, mice were positioned supine with limbs immobilized. After shaving and disinfecting the lower abdomen, a 1 cm midline incision was made along the linea alba to expose the ileocecal region. The junction of the ileocecal valve and distal cecum was ligated with sterile suture. Two transmural perforations were created at the midpoint between the ligation site and cecal tip using the 1 mL syringe, followed by gentle extrusion of cecal contents. The cecum was repositioned into the abdominal cavity, and the incision was closed in layers using absorbable sutures. All surgical procedures were conducted under strict aseptic conditions to minimize suffering. Anesthesia was induced and maintained with isoflurane to ensure a pain-free state during the operation. Following the procedure, postoperative analgesia (e.g., buprenorphine) was administered subcutaneously to alleviate pain. Body temperature was maintained throughout the surgery, and mice were monitored closely with appropriate supportive care (e.g., fluid supplementation) until they fully recovered from anesthesia.

Mice were randomly divided into five groups (n=8 per group):Control group (C): Received daily oral gavage of 0.5 mL phosphate-buffered saline (PBS) for 3 consecutive days prior to sham surgery, which involved identical procedures to the CLP operation except for cecal ligation and puncture. CLP group: Underwent CLP surgery after 3 days of daily PBS administration (0.5 mL/day via oral gavage). Lyc (Low-dose) + CLP group: Administered Lyc (10 mg/kg/day) via oral gavage for 3 days before CLP surgery. Lyc (Medium-dose) + CLP group: Treated with Lyc (20 mg/kg/day) orally for 3 days preceding CLP. Lyc (High-dose) + CLP group: Received Lyc (40 mg/kg/day) by oral gavage for 3 days prior to CLP induction.

Mice were subjected to cecal ligation and puncture (CLP) under isoflurane anesthesia. At 24 hours post-surgery, euthanasia was performed via cardiac carbon dioxide injection under continuous isoflurane anesthesia. The activity of the

mice was observed every four hours, and each observation lasted for 15 minutes, when animals exhibiting clinical signs of distress (including lethargy, convulsions, > 20% weight loss, inability to access food/water) were euthanizedor other adverse reactions) were promptly euthanized under isoflurane anesthesia when such manifestations were observed, mice were euthanized for biological sampling. Abdominal aorta blood was collected and allowed to clot at 4°C for 2 h, followed by centrifugation at 3,000 × g for 20 min to isolate serum. Serum aliquots were stored at −80°C for subsequent analysis. Renal tissues were dissected and divided into two portions: one was immersion-fixed in 4% paraformaldehyde (PFA) for 24 h at 4°C for histopathological staining, while the other was snap-frozen in liquid nitrogen and preserved at −80°C for molecular studies.

## Hematoxylin-eosin (HE) staining

For hematoxylin and eosin (H&E) staining, sections were dewaxed in xylene I (10–15 min) and xylene II (10–15 min), rehydrated through a graded ethanol series (absolute ethanol twice for 1 min, 95% ethanol for 1 min, 85% ethanol for 1 min), and rinsed with tap water (4×). Hematoxylin staining (5–10 min) was followed by tap water rinses (3×), differentiation in 2% hydrochloric acid-alcohol (30 sec), and bluing in tap water (12 h). Sections were counterstained with 0.5% eosin (1.5 min), rinsed with tap water (2×), dehydrated through graded ethanol (85% for 20 sec, 95% for 20 sec, absolute ethanol I and II for 10 min each), cleared in xylene I and II (10 min each), and mounted with neutral balsam for microscopic analysis.

## Immunohistochemical(IHC) staining

Following dewaxing as described above, paraffin sections underwent antigen retrieval by immersion in antigen retrieval buffer heated to 98°C in a water bath for 10–15 min, followed by cooling to room temperature (20–30 min). Sections were washed twice with PBS (30 min each). Endogenous peroxidase activity was blocked using 3% hydrogen peroxide for 10–15 min at room temperature, followed by three PBS washes (5 min each). Permeabilization was performed with 0.3% Triton X-100 for 30 min, and nonspecific binding was blocked with 10% normal goat serum for 1 h. Primary antibody (diluted 1:50 in PBS) was applied to sections and incubated overnight at 4°C. After three PBS washes (5 min each), biotinylated goat anti-rabbit secondary antibody was added and incubated for 2 h at room temperature on a shaker. Sections were washed again (PBS, 3 × 5 min), incubated with HRP-conjugated streptavidin for 1 h at room temperature, and developed using a DAB substrate kit. Staining was terminated by distilled water once brown coloration emerged. Sections were dehydrated, cleared in xylene, and neutral resin. For quantification, three sections per mouse (n = 3 mice/group) were analyzed. Positive cells were counted in five consecutive high-power fields (HPF, 400 × magnification) per section, and results were expressed as mean ± SD.

## ELISA assays

Renal tissue samples stored at −80°C were thawed on ice, rinsed with ice-cold PBS, blotted dry, and weighed (100 mg). Tissues were minced into fragments using sterile scissors and homogenized in 0.9 mL ice-cold PBS using a mechanical homogenizer. The homogenate was centrifuged at 3,000 × g for 15 min at 4°C, and the supernatant was collected for subsequent analysis.Renal homogenates were assayed for TNF-α, IL-1β, and IL-6 (serum) as well as malondialdehyde (MDA), superoxide dismutase (SOD), glutathione (GSH), and myeloperoxidase (MPO) (renal tissue) using commercially available ELISA kits according to the manufacturer's protocols. Briefly, the assays were performed as follows: standards and samples were added to the antibody-precoated wells and incubated. After washing, biotin-conjugated detection antibodies were added, followed by incubation with streptavidin-HRP. The chromogenic substrate TMB was then added, and the reaction was stopped with sulfuric acid solution. The absorbance at 450 nm was measured using a microplate reader. The concentrations of inflammatory cytokines and oxidative stress markers were determined by interpolating from the respective standard curves and normalized to the total protein concentration of each sample as determined by the BCA assay.

## Biochemical assays

Following a 4-hour clotting period at 4°C, whole blood samples were processed by centrifugation (3,000 × g, 20 min, 4°C) to isolate serum. Serum biomarkers of hepatic injury (AST, ALT, LDH, ALP) and renal dysfunction (BUN, CREA) were quantified using an automated biochemical analyzer.

## Western blotting

Renal tissue specimens stored at −80°C were thawed on ice, rinsed with ice-cold PBS, blotted dry, and weighed (100 mg). Tissues were homogenized in 0.9 mL of RIPA lysis buffer (containing 1% protease and phosphatase inhibitors) using a mechanical homogenizer. The homogenate was centrifuged at 12,000 × g for 15 min at 4°C, and the supernatant was collected. Total protein concentration was quantified via BCA assay, and samples were mixed with 6 × Laemmli loading buffer (1:5 ratio), followed by denaturation at 98°C for 10 min in a heating block. Proteins (20 μg per lane) were separated on 10% SDS-PAGE gels (80 V for stacking, 120 V for resolution) and transferred to PVDF membranes (300 mA, 90 min). Membranes were blocked with 5% non-fat milk in TBST for 2 h at room temperature, then incubated overnight at 4°C with primary antibodies against β-actin (monoclonal, mouse, 1:1000, Cat# ab8226, Abcam), AKT (monoclonal, mouse, 1:2000, Cat# 4691, CST), Bax (monoclonal, mouse, 1:1000, Cat# 5023, CST), Bcl-2 (monoclonal, mouse, 1:1000, Cat# 3498, CST), SOD1 (monoclonal, mouse, 1:1000, Cat# 4266, CST), cleaved caspase-3 (polyclonal, mouse, 1:500, Cat# 9664, CST), Nrf-2 (monoclonal, mouse, 1:1000, Cat# 12721, CST), PI3K (monoclonal, mouse, 1:1000, Cat# 4257, CST), p-AKT (monoclonal, mouse, 1:2000, Cat# 4060, CST), and p-PI3K (polyclonal, mouse, 1:1000, Cat# 4228, CST). After three TBST washes (10 min each), membranes were incubated with HRP-conjugated secondary antibodies (1:5000, Cat# 7074, CST) for 2 h at room temperature. Protein bands were visualized using an enhanced chemiluminescence (ECL) substrate (Cat# 34580, Thermo Fisher) and imaged with a ChemiDoc MP System (Bio-Rad). Densitometric analysis was performed using Image Lab software (v6.1).

## Cell culture

The human renal epithelial cell line HK-2 (purchased from the American Type Culture Collection, CRL-2190) was maintained in Dulbecco's Modified Eagle Medium (DMEM) supplemented with 10% fetal bovine serum (FBS), 0.1 U/mL bovine insulin, 50 U/mL penicillin, and 50 U/mL streptomycin at 37°C under a humidified 5% $CO_2$ atmosphere. Culture medium was refreshed every 48 hours. Upon reaching 90% confluency, cells were detached using 0.25% trypsin-EDTA (0.02%) for 3–5 min at 37°C, neutralized with complete medium, and centrifuged (200 × g, 5 min). Cells were either subcultured at a 1:3 ratio or cryopreserved in freezing medium (90% FBS, 10% DMSO) for subsequent experiments.

## Cell viability assay

HK-2 cells in logarithmic growth phase were seeded into 96-well plates ($1 \times 10^4$ cells/well, 100 μL medium/well) and allowed to adhere overnight under standard culture conditions (37°C, 5% $CO_2$). After complete attachment, cells were treated with serially diluted Lyc at designated concentrations and incubated for 24 h. Subsequently, 10 μL of CCK-8 reagent (10% v/v in serum-free DMEM) was added to each well, followed by 1 h incubation at 37°C. Absorbance was measured at 450 nm using a microplate reader (Model Synergy H1, BioTek),. All assays were performed in triplicate, and data were normalized to untreated control wells.

## In vitro inflammatory and oxidative stress models

HK2 cells at 80% confluency were preconditioned by replacing standard culture medium with serum-free DMEM, followed by stimulation with 1 μg/mL lipopolysaccharide (LPS) or 200 μM hydrogen peroxide ($H_2O_2$) for 24 h under standard conditions (37°C, 5% $CO_2$) to establish inflammatory or oxidative stress models. Post-stimulation, cells were treated with graded concentrations of Lyc for an additional 24 h. Cell viability was detected using the CCK-8 kit.

## Reactive oxygen species (ROS) assay

HK2 cells seeded in 6-well plates ($2 \times 10^5$ cells/well) were treated with Lyc, LPS (1 μg/mL), or $H_2O_2$ (200 μM) for 24 h. Following treatment, the supernatant was aspirated, and cells were incubated with 1 mL of 10 μM 2',7'-dichlorodihydrofluorescein diacetate (DCFH-DA) in serum-free DMEM at 37°C for 20 min in the dark. Cells were washed three times with ice-cold PBS to remove excess probe. Fluorescence imaging was performed using an inverted fluorescence microscope (Olympus IX73) equipped with a FITC filter (excitation/emission: 488/525 nm).

## Immunofluorescence staining

HK2 cells cultured on glass-bottom dishes were treated with 0–40 μM Lyc for 24 h to assess its modulatory effects on intracellular protein signaling. Cells were fixed with 4% paraformaldehyde (PFA) in PBS for 30 min at room temperature, permeabilized with 0.1% Triton X-100 on ice for 15 min, and blocked with 3% bovine serum albumin (BSA) for 1 h at 25°C. Primary antibody against phosphorylated PI3K (p-PI3K, 1:100 dilution, Cat# 4228, CST) was applied overnight at 4°C. After three PBS washes, cells were incubated with Alexa Fluor 488-conjugated secondary antibody (1:1,000, Cat# A-11008, Invitrogen) for 1 h at 37°C in the dark. Nuclei were counterstained with DAPI (1 μg/mL) in antifade mounting medium (Cat# H-1200, Vector Labs) for 15 min. Fluorescence signals were captured using a laser scanning confocal microscope (Olympus SpinS R) with excitation/emission settings of 488/519 nm (p-PI3K) and 405/461 nm (DAPI).

## Mitochondrial membrane potential (MMP) assay

HK2 cells seeded in 6-well plates ($5 \times 10^5$ cells/well) were treated with Lyc (0–40 μM), LPS (1 μg/mL), or $H_2O_2$ (200 μM) for 24 h. Post-treatment, the supernatant was aspirated, and cells were incubated with 1 mL JC-1 staining working solution (5 μg/mL in serum-free DMEM) at 37°C for 20 min under 5% $CO_2$. Concurrently, 1X JC-1 staining buffer was prepared by diluting 5X stock solution with distilled water (1:4 ratio) and chilled on ice. Following incubation, cells were washed twice with ice-cold 1X JC-1 buffer and replenished with 2 mL complete medium (containing 10% FBS and phenol red). Fluorescence signals were captured using a confocal microscope (Zeiss LSM 9000) with dual-excitation wavelengths: 488 nm (JC-1 aggregates, red emission: 590 nm) and 514 nm (JC-1 monomers, green emission: 529 nm). The ratio of aggregate-to-monomer fluorescence intensity was quantified using ZEN software (v3.0) to evaluate mitochondrial depolarization, with ≥5 fields analyzed per condition.ΔΨm values are expressed as fold change relative to the control group.

## Flow cytometry

Cellular apoptosis in HK2 cells was evaluated via dual-staining flow cytometry (Annexin V-FITC/PI assay, Cat# KGA107, KeyGEN BioTECH). Post-treatment, adherent cells were enzymatically detached using 0.25% trypsin-EDTA-free solution (Cat# T4049, Sigma-Aldrich), pelleted by centrifugation ($300 \times g$, 5 min), and washed twice with PBS (pH 7.4). Cell suspensions ($1 \times 10^6$ cells/mL) were resuspended in 100 μL Annexin V binding buffer and sequentially stained with 5 μL Annexin V-FITC and 5 μL PI under light-protected conditions. After 15 min incubation at 25°C, apoptotic populations were immediately analyzed on a BD FACSAria III flow cytometer (BD Biosciences), with compensation controls and gating strategies applied using FACSDiva software (v8.0.1).

## ER assays

Following experimental treatments, cellular specimens underwent three PBS rinses and were subsequently exposed to dual-organelle fluorescent probes: ER-Tracker™ Red and MitoTracker™ Green (Cat# C1041M, Beyotime), reconstituted in pre-equilibrated serum-free DMEM (37°C). Cells were maintained in this staining solution for 15 min under physiological conditions (37°C, 5% $CO_2$), followed by two gentle washes with complete medium to minimize non-specific probe

retention. Live-cell imaging was performed immediately using an LSM 900 confocal microscope (Carl Zeiss) equipped with a climate-controlled chamber (37°C, 5% $CO_2$).

## Cell transfection

HK2 cells were plated in 6-well culture dishes at a density of $2 \times 10^5$ cells/well and allowed to adhere for 24 h in DMEM medium supplemented with 10% FBS. For genetic perturbation, Lipofectamine™ 3000 (Cat# L3000015, Thermo Fisher Scientific) was utilized to deliver either 50 nM PI3K-specific siRNA or 2 μg PI3K overexpression plasmid into cells, following the reverse transfection protocol. Post-transfection (48 h), cells were exposed to oxidative stressors—200 μM $H_2O_2$ or 1 μg/mL LPS—in the presence or absence of 40 μM lycopene.

## Statistical analysis

Unless otherwise specified, quantitative data are presented as mean ± standard deviation (SD). The normality of all data distributions was verified using the Shapiro–Wilk test. For comparisons between two groups, a two-tailed unpaired Student's t-test was used. Comparisons among multiple groups were performed using one-way analysis of variance (ANOVA), followed by Tukey's post hoc test for multiple comparisons. A P-value of less than 0.05 was considered statistically significant. All statistical analyses were carried out with GraphPad Prism software (version 8.0). The specific significance levels in the figures are denoted as follows: *$P < 0.05$, **$P < 0.01$ versus the CLP/LPS/$H_2O_2$ group; #$P < 0.05$, ##$P < 0.01$ versus the sham/control group.

## Results

### Protective effects of lycopene against CLP-induced renal damage

As shown in Fig 1A, lycopene, a lipophilic carotenoid, exhibits superior antioxidant activity attributed to its extensive conjugated diene system (11 linearly arranged double bonds), enabling efficient singlet oxygen quenching and free radical neutralization. This nonpolar, acyclic structure enhances lipid-phase solubility, optimizing oxidative stress reduction in hydrophobic cellular environments, thereby mitigating lipid peroxidation and associated pathologies.

As shown in Fig 1B, CLP-induced renal injury in mice was characterized by tubular epithelial vacuolization, interstitial inflammatory infiltration, and cast formation. Lycopene treatment (10, 20, 40 mg/kg) dose-dependently attenuated these pathological features. The trend of pathological recovery after injury was consistent with decreased AST, ALT, CREA and BUN, confirming structural and functional recovery linked to lycopene's antioxidant properties, shown in Fig 1C.

### Protective effects of lycopene against CLP-induced renal inflammation

As demonstrated in Fig 2A, CLP-induced systemic inflammation was associated with significantly elevated serum levels of proinflammatory cytokines and increased renal MPO activity compared to the control group ($p < 0.01$). Pretreatment with Lyc (10, 20, 40 mg/kg) dose-dependently suppressed these inflammatory markers, with the highest dose reducing TNF-α, IL-6, and IL-1β to 102.3 ± 12.88, 108.67 ± 22.12, and 198.66 ± 33.5 ng/gprot, respectively ($p < 0.001$), and decreasing MPO activity to 22.83 U/g tissue ($p < 0.01$), indicating reduced neutrophil infiltration.

### Protective effects of lycopene against CLP-induced renal oxidative stress

As depicted in Fig 2B, CLP challenge markedly exacerbated renal oxidative stress, evidenced by a 62.69% reduction in SOD activity (Control: 130.19 ± 12.19 vs. CLP: 48.5 ± 7.50 U/mg protein, $p < 0.01$), a 76.70% decline in GSH levels (Control: 10.30 ± 0.63 vs. CLP: 2.40 ± 0.43 μmol/g tissue, $p < 0.001$), and a 2.2-fold increase in MDA content (Control: 9.20 ± 1.27 vs. CLP: 20.27 ± 1.23 nmol/mg protein, $p < 0.01$). Lyc pretreatment (10, 20, 40 mg/kg) dose-dependently

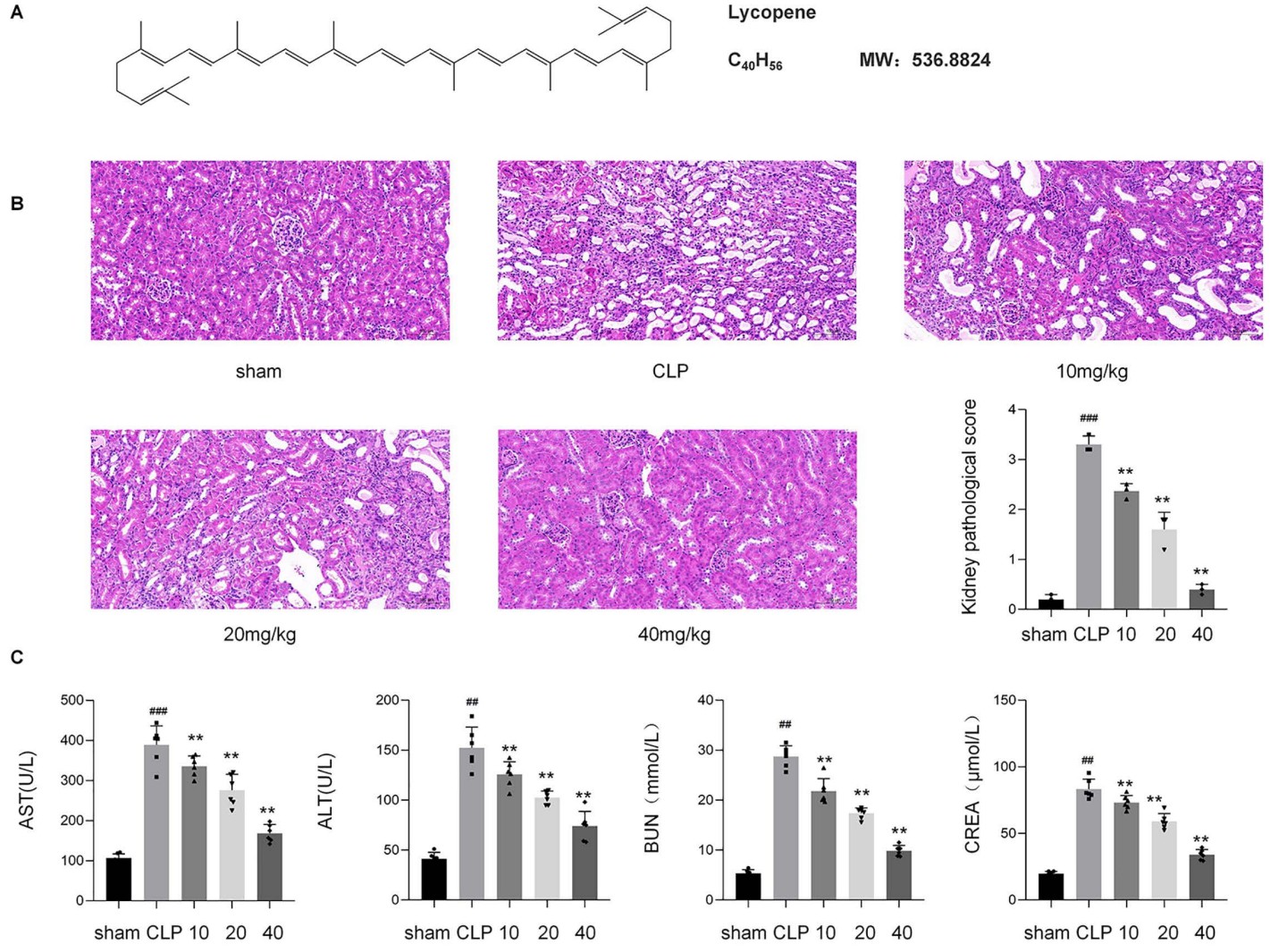

**Fig 1. Lycopene alleviates CLP-induced renal injury in mouse.**

restored antioxidant capacity, with the 40 mg/kg dose normalizing SOD (113.00±6.57 U/mg) and GSH (8.78±0.73 µmol/g) to near-control levels ($p < 0.05$ vs. CLP) while suppressing MDA accumulation to 13.08±0.95 nmol/mg protein ($p < 0.01$). These results underscore Lyc's efficacy in counteracting oxidative injury through redox homeostasis modulation.

## Protective effects of lycopene against I/R-induced renal apoptosis

As shown in the immunohistochemical results of Fig 2C, compared with the control group, the CLP group exhibited a marked increase in nuclear P53 protein expression. However, with the administration of increasing doses of lycopene, the positive rate of nuclear P53 staining was significantly reduced in a dose-dependent manner, suggesting that lycopene mitigates CLP-induced apoptotic responses in murine renal tissues through its dose-dependent regulatory effects.

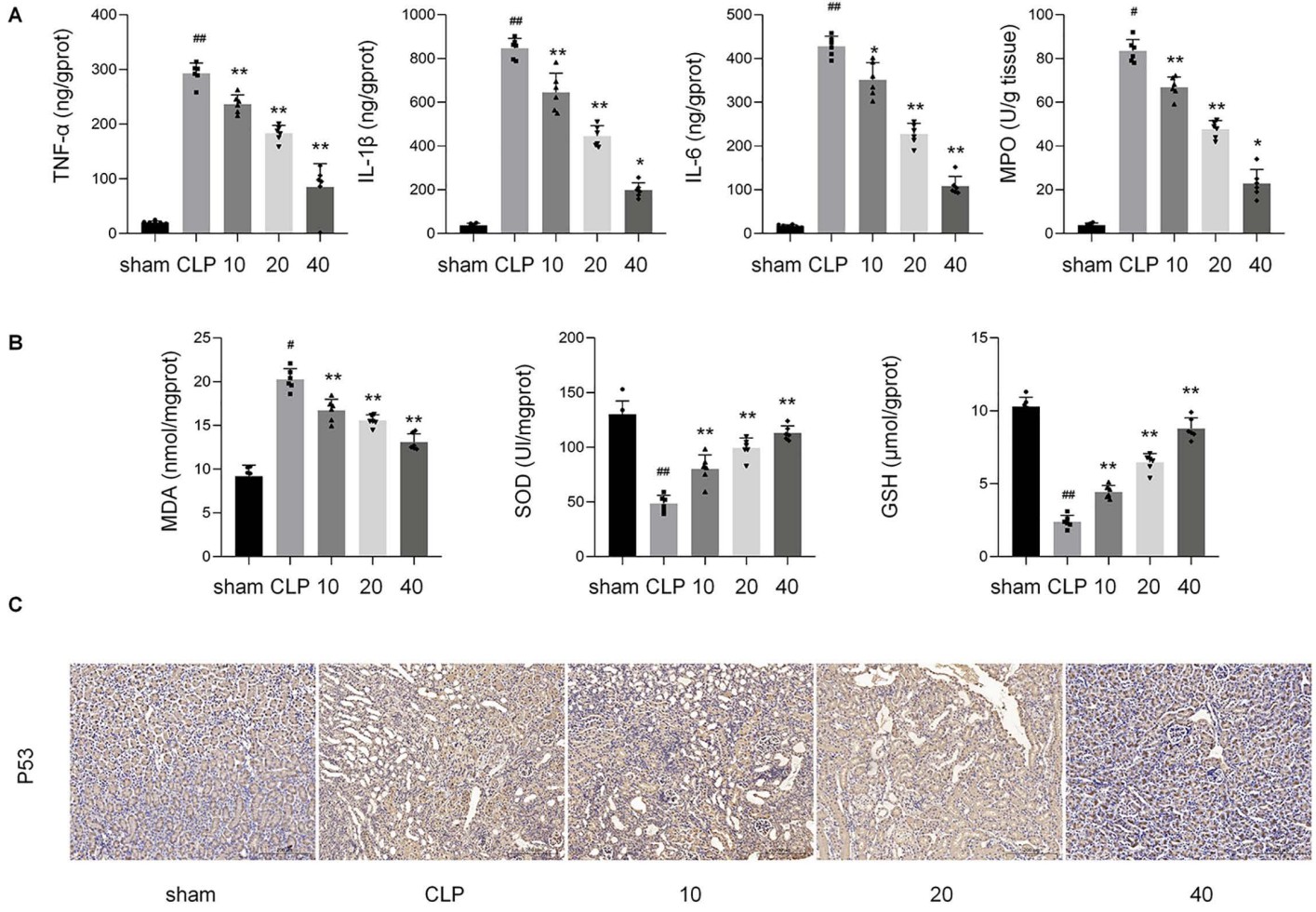

**Fig 2. Lycopene attenuates CLP-induced renal inflammation, oxidative stress, and apoptosis.**

## Lycopene inhibits ER stress, apoptosis and increases AKT/PI3K and antioxidant-related proteins in CLP-injured renal

As shown in Fig 3A, CLP-induced renal injury significantly increased phosphorylation levels of PI3K (p-PI3K: 3.1-fold vs. control, $p < 0.01$) and AKT (p-AKT: 2.9-fold, $p < 0.001$), along with diminished expression of antioxidant proteins Nrf-2 (0.3-fold, $p < 0.001$) and SOD1 (0.2-fold, $p < 0.01$), and anti-apoptotic Bcl-2 (0.1-fold, $p < 0.01$). Conversely, pro-apoptotic Bax (3.4-fold) and Cleaved-Caspase-3 (4.1-fold) were markedly upregulated ($p < 0.01$). Lycopene pretreatment (10, 20, 40 mg/kg) dose-dependently restored Nrf 2, SOD1, and Bcl-2/Bax to 86–98% of control levels ($p < 0.05$ vs. CLP) while suppressing p-PI3K, p-AKT, Cleaved-Caspase-3 and Bax by 67–78% ($p < 0.01$). Total PI3K and AKT remained unchanged, confirming modulation occurred at the phosphorylation level. These results highlight lycopene's role in suppressing the PI3K/AKT axis to mitigate oxidative stress and apoptosis in septic kidneys.The immunohistochemical results of p-PI3K and Nrf 2 were consistent with those of western bolt shoued in Fig 3B.

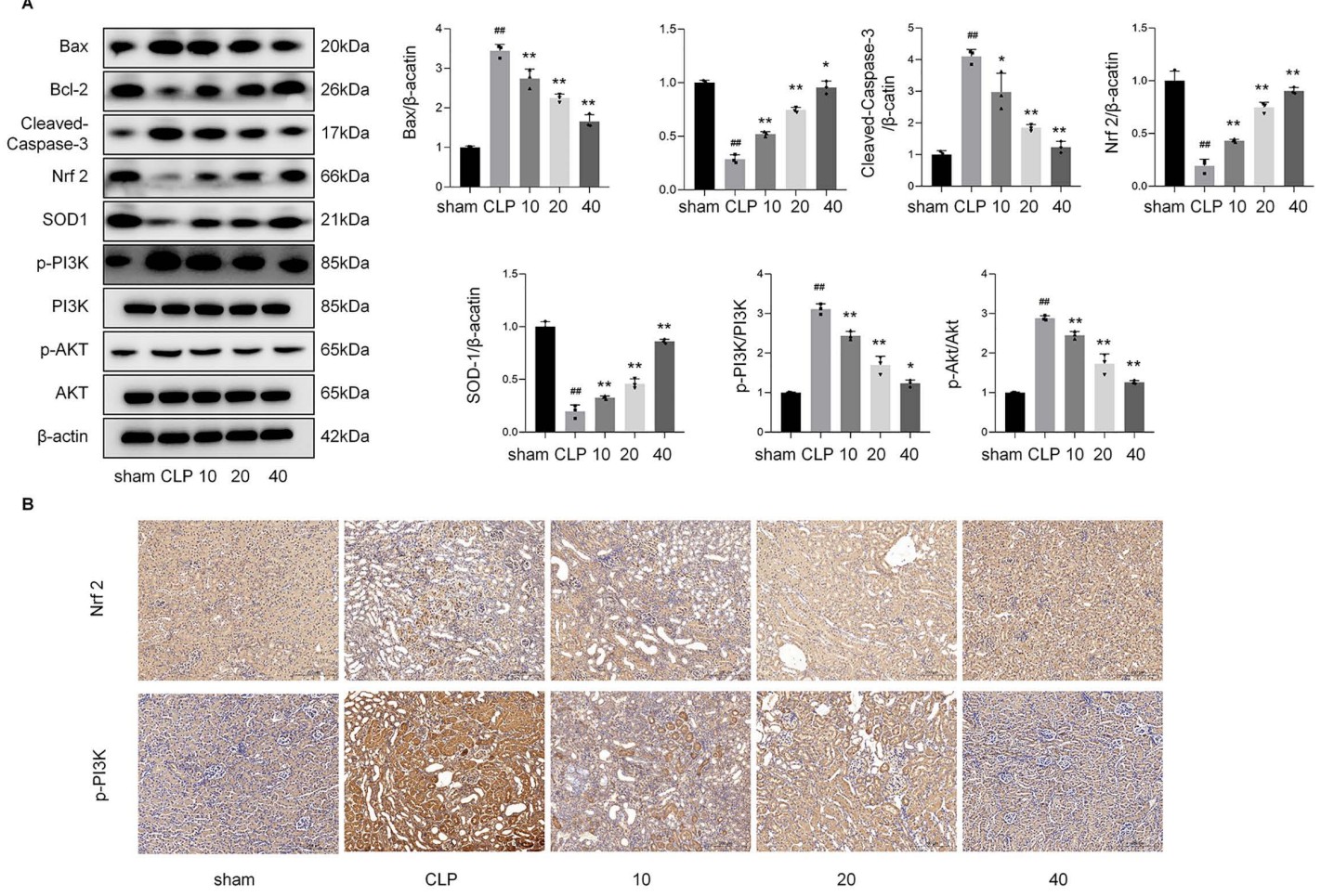

**Fig 3. Lycopene alleviates renal injury by targeting PI3K/AKT signaling and restoring antioxidant/anti-apoptotic balance.**

## Toxicity of lycopene in mouse

To evaluate the in vivo safety of lycopene, mice received intraperitoneal administration of a high-concentration lycopene formulation (40 mg/kg), followed by systemic physiological and histopathological assessments. Post-treatment analysis (10 days) revealed preserved tissue integrity in major organs (heart, liver, spleen, lung, kidney), with no detectable histo-pathological lesions or inflammatory infiltrates (Fig 4A). Serum biochemical indices, including hepatic (ALT, AST) and renal functional markers (BUN, CREA), remained within physiological ranges, showing no statistically significant alterations compared to baseline ($p > 0.05$) (Fig 4B). These data collectively affirm the biocompatibility and absence of acute toxicity associated with lycopene at the tested dosage.

## Protective effects of lycopene on cell viability

As shown in Fig 5A, CCK-8 assays revealed no significant cytotoxicity of Lyc (≤80 μM, 24 h) on normal HK2 cells, with cell viability maintained at 1.01±0.06 of untreated controls ($p > 0.05$). In contrast, LPS challenge markedly reduced HK2 cell viability to 0.64±0.03 of baseline ($p < 0.001$ vs. control), while pretreatment with Lyc (10, 20, 40 μM, 24 h) dose-dependently restored viability to 0.81±0.04, 0.89±0.03, and 0.95±0.01, respectively ($p < 0.05$–0.001 vs. LPS group)

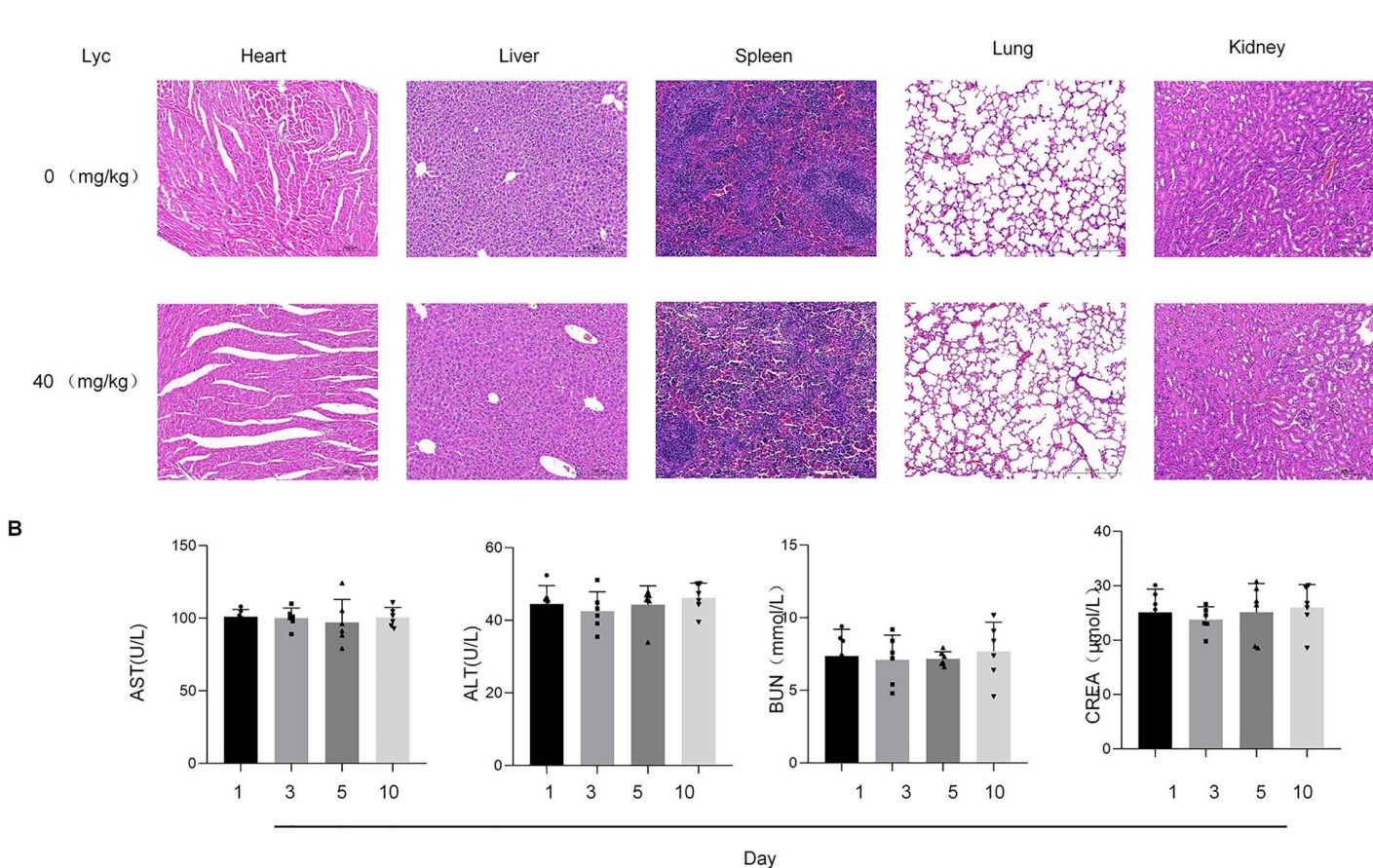

**Fig 4. Lycopene demonstrates no acute toxicity and maintains systemic biocompatibility in mice.**

(Fig 5B). Similarly, in an H₂O₂-induced oxidative injury model (200 μM), Lyc (20–40 μM) significantly attenuated cytotoxicity, increasing HK2 cell survival by 70–93% shown in Fig 5C ($p < 0.01$ vs. $H_2O_2$ group). These findings demonstrate lycopene's dual protective roles against inflammatory and oxidative insults in renal epithelial cells.

## The ability of lycopene to clear ROS

As demonstrated in Fig 5D and 5E, LPS and $H_2O_2$ stimulation triggered robust intracellular ROS generation shown in fluorescence intensity, compared to untreated controls. Lycopene pretreatment (10, 20, 40 μM) dose-dependently suppressed ROS overproduction, reducing fluorescence signals by 10%, 33%, and 74% in LPS-treated cells ($p < 0.001$), and by 40%, 55%, and 86% in $H_2O_2$-exposed cells ($p < 0.05$–0.001 vs. injury groups). These results quantitatively confirm lycopene's capacity to mitigate oxidative stress across distinct injury paradigms, consistent with its role as a potent free radical scavenger.

## Protective effects of lycopene against LPS and H₂O₂-induced ER stress in vitro

As illustrated in Fig 5F–I, both LPS and $H_2O_2$ challenge induced severe ultrastructural damage to the endoplasmic reticulum (ER) and mitochondria, showed a great reduction in ER-tracker Red fluorescence and mitochondrial green fluorescence,

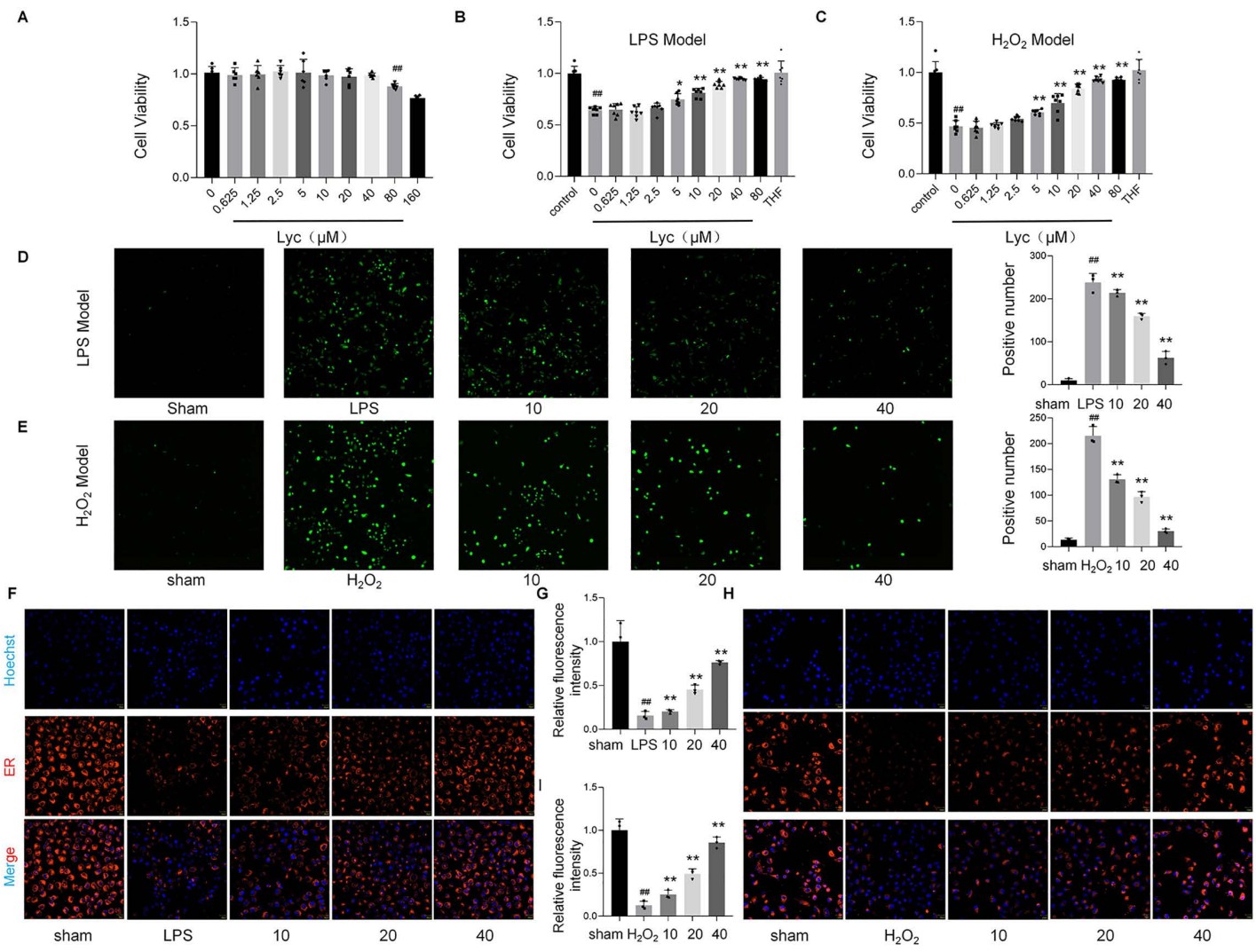

**Fig 5. Lycopene protects renal epithelial cells from inflammatory/oxidative injury by enhancing viability, scavenging ROS, and preserving organelle integrity.**

respectively, compared to the control group. Lycopene treatment (40 μM) significantly restored ER and mitochondrial fluorescence intensities to 76% ($p < 0.01$) and 86% ($p < 0.05$) of baseline levels, indicating structural and functional recovery of these organelles. Quantitative colocalization analysis further confirmed attenuated ER-mitochondria miscommunication. These findings underscore lycopene's role in maintaining organelle homeostasis under oxidative and inflammatory stress.

## Protective effects of lycopene against LPS and $H_2O_2$-induced apoptosis in vitro

Flow cytometry analysis revealed a pronounced increase in apoptotic cell populations in LPS- (42.15% ± 2.9%, $p < 0.001$) and $H_2O_2$-treated groups (55.7% ± 1.1%, $p < 0.001$) compared to controls (2.4% ± 0.62% and 2.0% ± 0.39%), while lycopene pretreatment (10–40 μM) dose-dependently reduced apoptosis to 27.8%–82.7% ($p < 0.01$–0.001 vs. injury groups) (Fig 6A, B). Consistent with these findings, mitochondrial membrane potential (ΔΨm) assays demonstrated that the LPS and $H_2O_2$ groups show a significant decrease in aggregates/monomer ratios (JC-1 staining) (0.37 ± 0.18 and 0.21 ± 0.11) compared

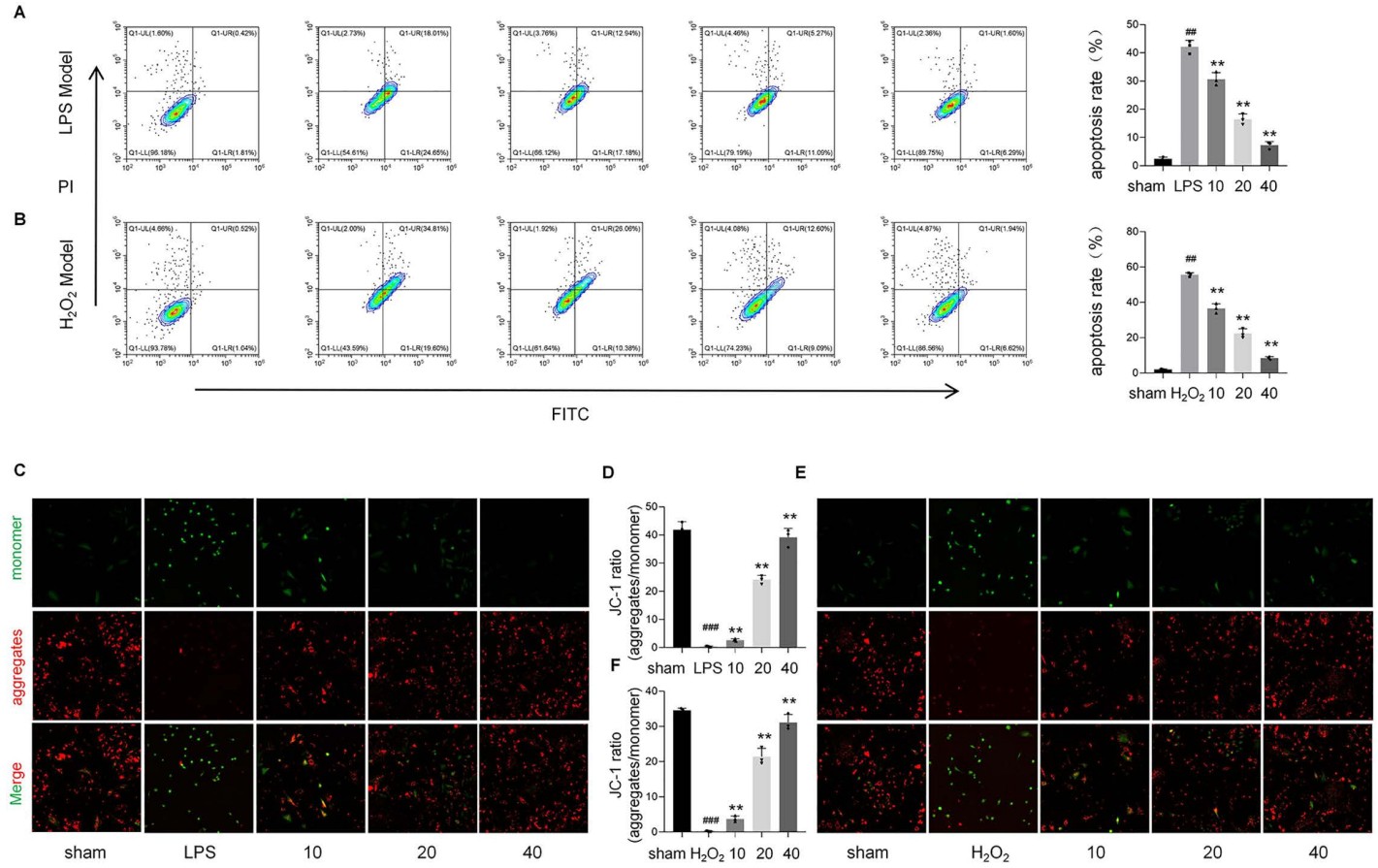

**Fig 6. Lycopene suppresses LPS/H₂O₂-induced apoptosis in vitro by stabilizing mitochondrial integrity and regulating intrinsic pathways.**

with the normal group (41.87±2.82 and 34.57±0.61) ($p < 0.001$), reflecting early mitochondrial depolarization. Lycopene restored ΔΨm ratios to 39.17±3.15 and 31.10±2.28, respectively ($p < 0.05$–$0.001$) (Fig 6C–I). These data mechanistically link lycopene's anti-apoptotic effects to mitochondrial stability and intrinsic apoptosis pathway regulation.

### Role of PI3K/AKT/Nrf-2 signaling pathway in lycopene-exerted protection

Western blot analysis in two distinct cellular models revealed consistent expression trends of p-PI3K, p-AKT, Nrf-2, SOD1, Bax, Cleaved Caspase-3, and Bcl-2 with those observed in animal tissues. Notably, the total protein levels of PI3K and AKT remained unaltered across all experimental groups (Fig 7A, B). Immunofluorescence staining of key proteins further corroborated these findings. The model group exhibited markedly increased fluorescence intensity, whereas lycopene treatment dose-dependently reduced fluorescence signals, suggesting an inhibition effect on protein expression (Fig 7C, D).

To investigate the correlation between lycopene-induced regulation and p-PI3K activity, HK2 cells were transfected with PI3K siRNA or cDNA. Transfection with PI3K siRNA significantly suppressed the expression of p-PI3K, p-AKT, and Bax levels, while upregulating Nrf-2, SOD1, and Bcl-2 protein expression. Co-treatment of lycopene and siRNA demonstrated consistent results with siRNA, suggesting that si-PI3K has a partial synergistic effect with lycopene (Fig 8A, B). Conversely, PI3K cDNA transfection can significantly increase the expression levels of p-PI3K, pro-apoptotic marker (Bax) proteins, and simultaneously reduce the expression levels of antioxidant proteins/anti-apoptotic proteins

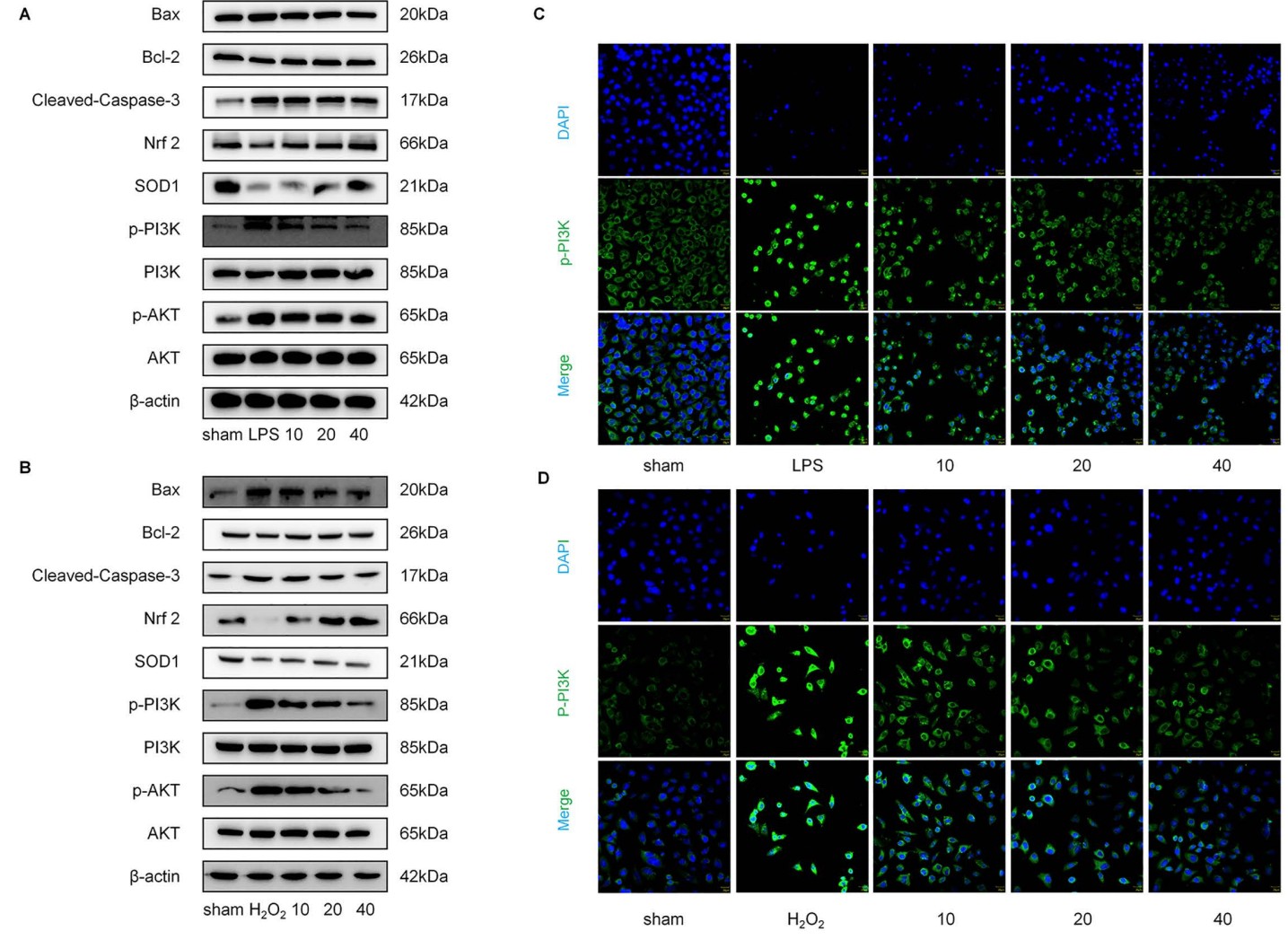

**Fig 7. Lycopene modulates the PI3K/AKT-Nrf-2 pathway to exert antioxidant and anti-apoptotic effects.**

(Nrf-2, SOD1, Bcl-2) proteins. However, the co-incubation of lycopene and PI3K cDNA nullified the cell protective effect of lycopene alone, suggesting that the antioxidant and anti-apoptotic abilities of lycopene may be achieved by inhibiting p-PI3K (Fig 8C, D).

These findings collectively demonstrate that lycopene exerts antioxidant and anti-apoptotic effects in cellular models of acute kidney injury, potentially mediated through modulation of the PI3K/AKT-Nrf-2 signaling pathway. The cytoprotective properties of lycopene appear dependent on its inhibitory action against p-PI3K, as evidenced by the abrogation of therapeutic efficacy upon PI3K overexpression and phenotypic replication through PI3K silencing experiments, which further corroborates this mechanistic pathway.

## Discussion

The present study demonstrates that lycopene exerts renoprotective effects in CLP-induced acute kidney injury through modulation of the PI3K/AKT signaling axis, as evidenced by comprehensive in vivo and in vitro experiments. Our findings

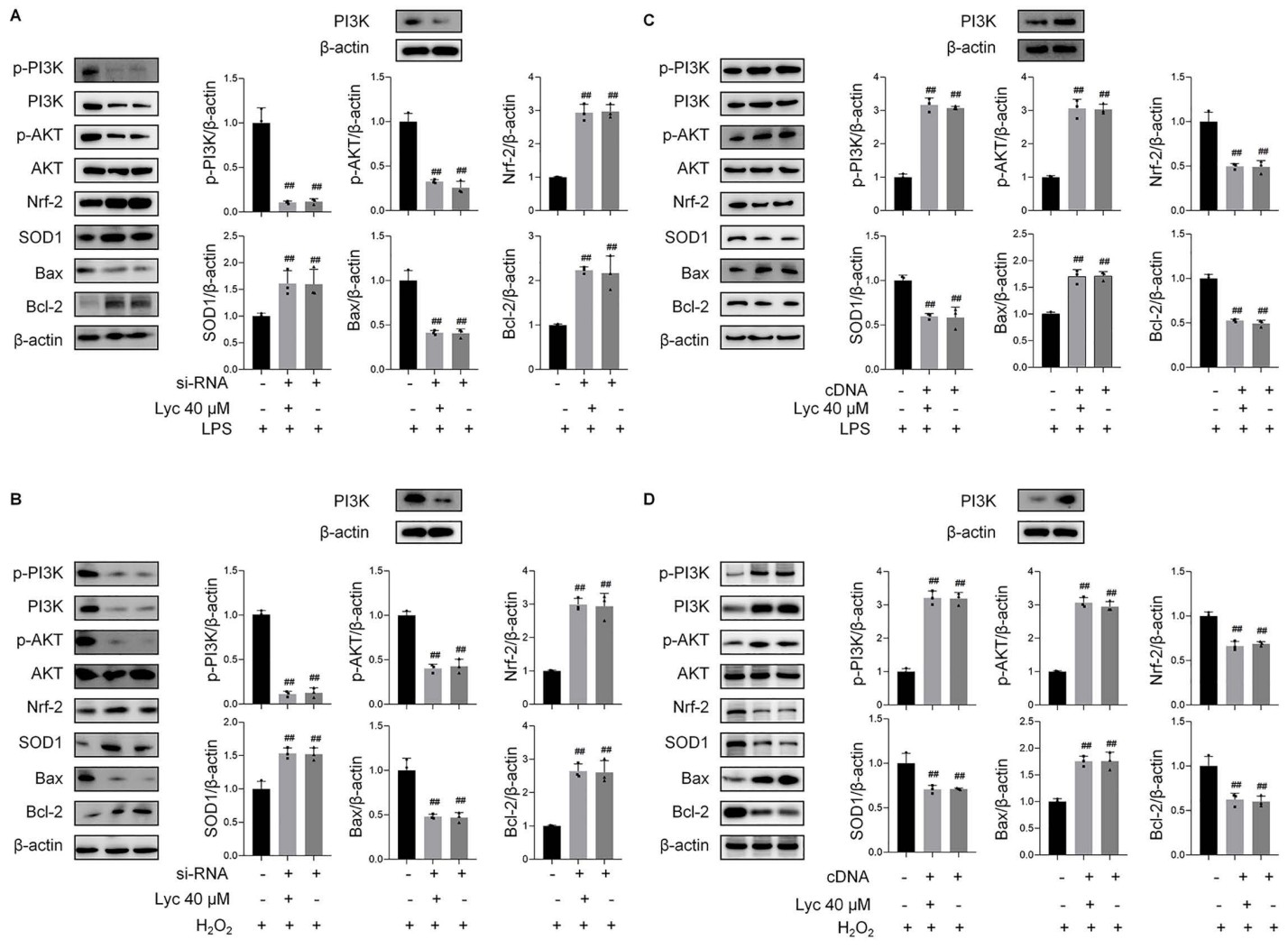

**Fig 8. Lycopene's cytoprotection depends on PI3K inhibition, validated by genetic manipulation.**

align with emerging evidence regarding the dual regulatory role of PI3K in cellular stress responses while providing novel insights into phytochemical-mediated pathway regulation [29]. The observed elevation of p-PI3K in CLP model group (Fig 3A), paradoxically associated with aggravated renal pathology, warrants careful interpretation. Previous studies have reported context-dependent PI3K activation patterns, where transient activation may initiate protective autophagy while sustained hyperactivation exacerbates oxidative stress and apoptosis [30,31]. In our sepsis-associated AKI model, the persistent PI3K phosphorylation likely reflects maladaptive signaling cascades triggered by polymicrobial infection, consistent with reports that PI3Kγ isoform overexpression exacerbates sepsis-induced organ dysfunction through neutrophil hyperactivation [32,33]. The dose-dependent suppression of p-PI3K by lycopene accompanied by Nrf 2 pathway restoration shown in Figs 3A and 7A, B, suggests a sophisticated regulatory mechanism that differs from direct PI3K inhibitors, potentially explaining the superior safety profile observed in our toxicity assessments [34,35].

In the present study, the assessment of oxidative stress was conducted using a multi-faceted strategy. The direct measurement of intracellular reactive oxygen species (ROS) generation via the DCFH-DA fluorescent probe confirmed

a significant oxidative burst in our disease model, which was remarkably attenuated by lycopene treatment (Fig 5D–F). More importantly, to substantiate the presence of sustained endogenous oxidative damage, we quantified key biochemical markers in renal tissues. The observed elevation in malondialdehyde (MDA), a reliable indicator of lipid peroxidation, coupled with the significant depletion of the primary antioxidants superoxide dismutase (SOD) and glutathione (GSH), provides compelling and objective evidence that the disease state induces a profound imbalance of the redox systemsm, as presented in Fig 2B. Notably, lycopene administration effectively reversed these alterations, significantly lowering MDA content while restoring SOD and GSH levels. This coordinated amelioration of both oxidative damage and antioxidant defense by lycopene substantiates its potent redox-modulating capacity in the context of septic renal injury, likely through the activation of the Nrf 2 signaling pathway as demonstrated in our results.

Notably, the multi-model validation strategy significantly strengthens our conclusions. The concordance between CLP-induced AKI and dual LPS/$H_2O_2$ cellular models in demonstrating PI3K pathway dysregulation supports the fundamental role of this pathway in sepsis-related renal injury (Fig 7A–D). Our findings extend previous reports of lycopene's antioxidant properties by [36,37] mechanistically linking its therapeutic effects to PI3K/AKT modulation. The reduction in p-PI3K immunoreactivity at the highest lycopene dose (40 mg/kg) correlates with decrease in tubular injury scores, suggesting pathway centrality in pathology progression (Fig 1). Importantly, the genetic verification using PI3K silencing and overexpression provides causal evidence surpassing conventional pharmacological inhibition approaches, addressing limitations in previous phytochemical studies (Fig 8).

The temporal dynamics of PI3K activation may explain its "double-edged sword" characteristics [38,39]. Early-phase PI3K activation in AKI potentially mediates compensatory cytoprotective responses, while prolonged activation (as in our 24h CLP model) likely shifts toward promoting inflammatory and apoptotic pathways [40]. This aligns with clinical observations of worsened outcomes in patients with sustained PI3K/AKT/mTOR pathway activation [41–43]. Lycopene's ability to normalize rather than completely abolish p-PI3K levels suggests a homeostatic regulatory effect, potentially avoiding over-suppression-related complications seen with synthetic kinase inhibitors.

Our study advances current knowledge in three key aspects: First, it establishes PI3K as a critical nodal point in sepsis-associated AKI pathophysiology through orthogonal in vivo and in vitro evidence. Second, it identifies lycopene as a natural PI3K modulator with unique temporal and spatial regulatory characteristics. Third, the comprehensive safety assessment from cellular to organismal levels (80μM in vitro and 40 mg/kg in vivo) provides crucial preclinical data for therapeutic development (Fig 4). However, certain limitations merit consideration. The study focused on acute phase responses within 24h post-CLP, whereas chronic PI3K modulation effects require further investigation. Additionally, while we confirmed PI3K as the primary target, potential crosstalk with Nrf-2 and p53 pathways warrants deeper exploration through multi-omics approaches.

Our IHC analysis revealed a significant elevation in nuclear p53 protein levels in the CLP group, which was effectively mitigated by lycopene treatment (Fig 2C). This finding suggests a potential cross-talk between the PI3K/AKT/Nrf-2 axis and p53 signaling in our model. Given that p53 is a well-established sensor of cellular stress, its activation in sepsis-induced renal injury is likely a downstream consequence of the intense oxidative damage and inflammatory response. The antioxidant properties of lycopene, primarily mediated through the Nrf-2 pathway, may thus indirectly suppress p53 activation by alleviating the genotoxic and oxidative stress that triggers it. Alternatively, we cannot rule out the possibility that lycopene directly modulates p53 activity through other mechanisms. Elucidating the precise hierarchy between these pathways—whether p53 acts predominantly downstream of Nrf-2-governed redox homeostasis or is independently targeted by lycopene—warrants further investigation. Nonetheless, the observed modulation of p53 underscores the multifaceted protective role of lycopene, which appears to concurrently engage both cytoprotective (Nrf 2) and genotoxic stress (p53) response pathways.

While our findings demonstrate that lycopene attenuates sepsis-associated renal injury via the PI3K/AKT/Nrf 2 pathway in a CLP model, translating these preclinical results to clinical application requires careful consideration of human dosing

and bioavailability. The effective doses used in our study (10−40 mg/kg) align with the typical human equivalent doses of lycopene supplementation, which range from 10 to 15 mg daily in clinical studies for conditions involving oxidative stress and inflammation [44–46]. However, a key translational challenge remains its inherently low oral bioavailability. Notably, the bioavailability of lycopene from heat-processed tomato products (e.g., tomato paste) is significantly higher than from fresh tomatoes, which may partially explain the efficacy observed in epidemiological studies. Furthermore, recent advances in formulation strategies, particularly nanocarrier systems, have shown promise in enhancing lycopene's solubility, stability, and tissue distribution – potentially facilitating better targeting to renal tissues and more effective modulation of the PI3K/AKT/Nrf-2 pathway identified in our study. These considerations bridge our mechanistic findings with clinical translation and suggest that with appropriate formulation approaches, lycopene could represent a viable adjunctive therapy for sepsis-induced organ damage [44,47,48].

These findings carry significant translational implications. The dose-dependent efficacy with no observed toxicity at therapeutic doses positions lycopene as a promising adjuvant therapy for sepsis-induced AKI. Future studies should investigate optimal administration timing relative to PI3K activation phases and explore combination therapies with existing AKI treatments. The conserved protective mechanisms across species (mouse models and human HK2 cells) suggest broad clinical applicability, though pharmacokinetic studies are needed to establish human-equivalent dosing. Collectively, this work not only elucidates a novel mechanism of lycopene's renoprotection but also provides a framework for developing pathway-specific phytochemical interventions in critical illness.

## Supporting information

**S1 File. S1_raw_images.**
(PDF)

**S2 File. The ARRIVE guidelines 2.0 author checklist.**
(PDF)

## Acknowledgments

The authors thank Zhiyong XU, Yueli SHI, and Yun XU for their expert technical guidance in experimental procedures. We also acknowledge Yunqing Wang from the Core Facilities, the Fourth Affiliated Hospital of School of Medicine, and International School of Medicine, International Institutes of Medicine, Zhejiang University, for essential technical support.

## Author contributions

**Conceptualization:** Yuchao Sun.

**Data curation:** Lijun Zheng, Hui Zhang.

**Formal analysis:** Lijun Zheng, Hui Zhang.

**Funding acquisition:** Lijun Zheng.

**Methodology:** Lijun Zheng, Hui Zhang.

**Project administration:** Lijun Zheng.

**Resources:** Yuchao Sun.

**Software:** Hui Zhang.

**Supervision:** Hui Zhang, Yuchao Sun.

**Validation:** Yuchao Sun.

**Visualization:** Yuchao Sun.

**Writing – original draft:** Lijun Zheng, Hui Zhang.

**Writing – review & editing:** Yuchao Sun.

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
