## [Decision Letter · Decision Letter 0]

16 Jul 2025

Dear Dr. Sun,

Thank you for submitting your manuscript to PLOS ONE. After careful consideration, we feel that it has merit but does not fully meet PLOS ONE’s publication criteria as it currently stands. Therefore, we invite you to submit a revised version of the manuscript that addresses the points raised during the review process.

We look forward to receiving your revised manuscript.

Kind regards,

Ajit Prakash, PhD

Academic Editor

PLOS ONE

Journal Requirements:

4. We note that your Data Availability Statement is currently as follows: All relevant data are within the manuscript and its Supporting Information files

Supported by Foundation of Zhejiang Provincial Education Department, Y202353817

6. Please update your submission to use the PLOS LaTeX template. The template and more information on our requirements for LaTeX submissions can be found at http://journals.plos.org/plosone/s/latex.

Reviewers' comments:

Reviewer's Responses to Questions

**Comments to the Author**

1. Is the manuscript technically sound, and do the data support the conclusions?

Reviewer #1: Yes

Reviewer #2: Yes

2. Has the statistical analysis been performed appropriately and rigorously?

Reviewer #1: Yes

Reviewer #2: Yes

3. Have the authors made all data underlying the findings in their manuscript fully available?

Reviewer #1: Yes

Reviewer #2: Yes

4. Is the manuscript presented in an intelligible fashion and written in standard English?

Reviewer #1: Yes

Reviewer #2: Yes

Reviewer #1: Summary: Sepsis-induced acute kidney injury (AKI) is a serious medical condition with very limited treatment options. This study explores the potential of lycopene, a natural antioxidant found in tomatoes, to protect the kidneys in this context. The authors show that lycopene reduces inflammation, oxidative stress, and cell death in both mice and human kidney cells. The protective effect is shown to depend on modulation of the PI3K/AKT pathway, which is often overactive during sepsis, and by activating Nrf-2, a key protein that defends against cellular damage. The authors use genetic experiments to confirm that lycopene’s protective effects rely on blocking PI3K activity. These results suggest that lycopene may be developed as a potential therapy for sepsis-induced kidney injury.

2) Major Comments

1. The authors have not defined sepsis models with clarity and distinction. Multiple sub-section headings under the “Results” section refer to “I/R-induced renal injury” such as on page 20, 21 and 22 of the PDF file. However, in the text the authors have mostly discussed about cecal ligation and puncture (CLP) to induce sepsis, not ischemia-reperfusion (I/R). This needs clarity and scientific accuracy. Recommend authors to carefully evaluate the experimental model and correctly represent them throughout the manuscript to avoid misrepresenting the experimental design.

2. The authors state that they used Student’s t-test to compare two groups. However, as seen in Figures 1-3, the experiments involve five groups (Control, CLP, LYP 10, 20, 40 mg/kg). For multiple groups, one-way ANOVA followed by Tukey’s or Dunnett’s post-hoc test is considered as standard and more appropriate. The authors must update the statistical method section accordingly and clarify what tests were used per figure.

3. The IHC analysis of nuclear P53 (Figure 2C) is clearly presented, but its mechanistic role is not sufficiently integrated into the main pathway narrative (PI3K/AKT/Nrf-2).

In the Discussion, the authors note the need to explore the p53 pathway further. To improve the paper’s impact, briefly discuss whether P53 operates downstream of oxidative stress or is independently modulated by lycopene.

4. The authors suggest that lycopene has clinical potential, which is reasonable, but the claim would be stronger with context. Discussing known human dosing, bioavailability, or prior clinical uses of lycopene would help bridge the gap between preclinical models and therapeutic application. As it stands, the claim lacks detail and feels speculative. A short paragraph bridging this translational gap would improve the discussion.

Minor Comments

1. Typos and terminologies:

1.a Correct “downregulation the PI3K/Akt axis” to “downregulating the PI3K/Akt axis.” (Abstract, line 13))

1.b “apoptpsis” should be “apoptosis.” (Page 21; section title)

1.c Consistently use “H₂O₂” instead of “H202” or other variants.

1.d “lpy’s” should be “lycopene’s.” (Page 25, last para)

1.e Ensure spacing in section headers, e.g., “Immunohistochemical (IHC) Staining.” (Page 14, Methods section)

1.f “Elisa assays” should be “ELISA assays” (Page 15)

2. Figure Interpretation and clarity:

2.a Figure 3A: Clarify whether Bcl-2/Bax values refer to protein expression individually or as a ratio. The current presentation is ambiguous.

2.b Figure 6C-I: The reported 100-fold difference in ΔΨm between control and treated groups is unusually high. Recheck to confirm if this is an error or explain the unit/normalization method to avoid confusion.

2.c Figure 7B: Label states “immunohistochemical staining” but Methods indicate immunofluorescence in HK2 cells. Please clarify whether these are tissue or cell-based images and update terminology accordingly.

3. Rephrasing Manuscript Title:

The title “Lycopene inhibits ER stress, apoptosis and increases AKT/PI3K and antioxidant-related proteins” is confusing as lycopene reduces phosphorylated (active) PI3K and AKT, it should be rephrased.

Suggestion: “Lycopene inhibits ER stress and apoptosis while modulating PI3K/AKT and enhancing antioxidant and anti-apoptotic proteins.”

Reviewer #2: The authors have done a rigorous research analysis of the oxidative stress and effect of lycopene on kidney injury. The study investigates the renoprotective effects of lycopene, a powerful antioxidant, in sepsis-induced acute kidney injury (AKI). Using murine and cellular models, they found lycopene to mitigate oxidative stress, inflammation, and apoptosis by modulating the PI3K/AKT/Nrf-2 signaling pathway. It dose-dependently restored renal function, reduced pro-apoptotic markers, and enhanced antioxidant defenses. The authors have used a combination of techniques to justify their studies. Though the manuscript encompasses a lot of details, there are some things that need to be addressed in the manuscript. Below are my comments on the manuscript:

1. There are details missing in the entire manuscript. The introduction could have started with a generic problem and then discussed in detail.

2. The details of mortality rates are missing. The authors say high mortality. Better to use stats from literature.

3. The authors have abruptly started using the abbreviated version for lycopene, without mentioning it in the manuscript. The details should be in the manuscript, which makes it comfortable for the reader.

4. Though the manufactures protocols are followed in most of the kit based assays, still the method should be briefly explained in the manuscript.

5. Some of the assays done have not been talked about in the introduction. It would be better to have an idea of what all the things will the article be focusing on like CCK-8, different markers. This could be a section in the intro.

6. How have the authors quantified the oxidative stress? Have they checked the endogenous oxidation in the disease state too?

7. The results have just been mentioned as seen in figures and not discussed in detail. It would be beneficial to have it that way with discussions.

**Do you want your identity to be public for this peer review?** For information about this choice, including consent withdrawal, please see our Privacy Policy

Reviewer #1: No

Reviewer #2: No

---

## [Author Response · Author response to Decision Letter 1]

30 Sep 2025

Journal Requirements

1.Please ensure that your manuscript meets PLOS ONE's style requirements, including those for file naming. The PLOS ONE style templates can be found at https://journals.plos.org/plosone/s/file?id=wjVg/PLOSOne_formatting_sample_main_body.pdf and https://journals.plos.org/plosone/s/file?id=ba62/PLOSOne_formatting_sample_title_authors_affiliations.pdf

Response:

We thank the editors and reviewers for their time and insightful comments on our manuscript. We have carefully considered all the feedback and have revised the manuscript accordingly. The changes made address every point raised, and we believe the manuscript has been significantly improved. We have also ensured that the manuscript now fully complies with PLOS ONE's style requirements, including those for file naming, figure formatting, and reference presentation.

2.PLOS ONE now requires that authors provide the original uncropped and unadjusted images underlying all blot or gel results reported in a submission’s figures or Supporting Information files. This policy and the journal’s other requirements for blot/gel reporting and figure preparation are described in detail at https://journals.plos.org/plosone/s/figures#loc-blot-and-gel-reporting-requirements and https://journals.plos.org/plosone/s/figures#loc-preparing-figures-from-image-files. When you submit your revised manuscript, please ensure that your figures adhere fully to these guidelines and provide the original underlying images for all blot or gel data reported in your submission. See the following link for instructions on providing the original image data: https://journals.plos.org/plosone/s/figures#loc-original-images-for-blots-and-gels.

Response:

We confirm that we have complied with PLOS ONE's policy for reporting blot/gel results. As required, we have provided the original, uncropped, and unadjusted images underlying all Western blot results reported in the figures. These images have been compiled and uploaded as a supporting information file labeled "S1_raw_images".

3.Your ethics statement should only appear in the Methods section of your manuscript. If your ethics statement is written in any section besides the Methods, please delete it from any other section.

Response:

We confirm that the ethics statement, including the approval number, is now present solely in the Methods section of the manuscript. Any duplicate statements have been removed from other sections of the manuscript to ensure full compliance with journal policy.

4.We note that your Data Availability Statement is currently as follows: All relevant data are within the manuscript and its Supporting Information files...

Response:

We confirm that our submission contains the minimal data set required to replicate all findings reported in our study. All underlying raw data (including the individual values behind means, statistical analyses, and data used to generate graphs) have been uploaded as Supporting Information files with the manuscript. Therefore, our Data Availability Statement remains accurate.

5.Thank you for stating the following financial disclosure:

Supported by Foundation of Zhejiang Provincial Education Department, Y202353817

Please state what role the funders took in the study. If the funders had no role, please state: "The funders had no role in study design, data collection and analysis, decision to publish, or preparation of the manuscript."...

Response:

Thank you for pointing this out. We have amended our funding statement to clarify that the grant was awarded to the first author. The updated statement in the manuscript reads:

"This work was supported by the Foundation of Zhejiang Provincial Education Department (Grant Number Y202353817) awarded to Lijun Zheng."

Additionally, as required, we confirm that the funders had no role in study design, data collection and analysis, decision to publish, or preparation of the manuscript.

6.Please update your submission to use the PLOS LaTeX template. The template and more information on our requirements for LaTeX submissions can be found at http://journals.plos.org/plosone/s/latex.

Response:

We thank the editor for this reminder. We have now updated our submission to fully comply with the PLOS ONE LaTeX template requirements. Our manuscript has been reformatted using the official PLOS LaTeX template, and all supporting files have been prepared according to the specified guidelines.

7.If the reviewer comments include a recommendation to cite specific previously published works, please review and evaluate these publications to determine whether they are relevant and should be cited. There is no requirement to cite these works unless the editor has indicated otherwise.

Response:

We confirm that the reviewers' comments did not include any recommendations to cite specific previously published works. Therefore, no additional citations have been added in response to reviewer feedback.

Reviewer 1 Comments to the Author

1. Major Comments

The authors have not defined sepsis models with clarity and distinction. Multiple sub-section headings under the “Results” section refer to “I/R-induced renal injury” such as on page 20, 21 and 22 of the PDF file. However, in the text the authors have mostly discussed about cecal ligation and puncture (CLP) to induce sepsis, not ischemia-reperfusion (I/R). This needs clarity and scientific accuracy. Recommend authors to carefully evaluate the experimental model and correctly represent them throughout the manuscript to avoid misrepresenting the experimental design.

Response:

We sincerely thank the reviewer for this exceptionally careful reading and for pointing out this critical error in our manuscript. The reviewer is absolutely correct. We mistakenly used the term "I/R-induced renal injury" in several sub-section headings and parts of the text, while the experiments were in fact conducted using the cecal ligation and puncture (CLP) model to induce sepsis-associated renal injury.

This was an oversight during the writing and revision process, and we apologize for the lack of clarity and scientific inaccuracy. We have now thoroughly corrected the entire manuscript to ensure it accurately reflects the CLP model used in our study. Specifically:

All incorrect sub-section headings (e.g., on pages 20, 21, and 22 of the previous PDF) and in-text mentions of "I/R" have been replaced with the accurate term "CLP-induced septic renal injury" or similar appropriate phrasing.

We have carefully checked the entire text to ensure consistency and scientific accuracy in describing our experimental model. We believe these corrections have resolved the issue and have prevented any misrepresentation of our experimental design.

2.The authors state that they used Student’s t-test to compare two groups. However, as seen in Figures 1-3, the experiments involve five groups (Control, CLP, LYP 10, 20, 40 mg/kg). For multiple groups, one-way ANOVA followed by Tukey’s or Dunnett’s post-hoc test is considered as standard and more appropriate. The authors must update the statistical method section accordingly and clarify what tests were used per figure.

Response:

We sincerely thank the reviewer for pointing out the inaccuracy in our statistical description. We apologize for this oversight; while we had indeed employed one-way ANOVA followed by Tukey’s post-hoc test for multiple group comparisons, we inadvertently described it incorrectly due to habitual wording. The Statistical Analysis section in the Methods has now been carefully revised to accurately reflect the correct statistical approach. The specific significance notations used in the figures are as follows: *P < 0.05, **P < 0.01 vs CLP/LPS/H2O₂group; #P < 0.05, ##P < 0.01 vs sham/Control group. We have ensured that all statistical reporting throughout the manuscript is now consistent and accurate.

3.The IHC analysis of nuclear P53 (Figure 2C) is clearly presented, but its mechanistic role is not sufficiently integrated into the main pathway narrative (PI3K/AKT/Nrf-2).

In the Discussion, the authors note the need to explore the p53 pathway further. To improve the paper’s impact, briefly discuss whether P53 operates downstream of oxidative stress or is independently modulated by lycopene.

Response:

We sincerely thank the reviewer for this insightful suggestion to better integrate our findings on p53 into the central mechanistic narrative of the paper. We agree that this discussion enhances the impact and coherence of our story.

As requested, we have now added a dedicated paragraph in the Discussion section to briefly elaborate on the potential role of p53 in relation to the PI3K/AKT/Nrf-2 pathway and lycopene's action. The added text discusses the likelihood that p53 activation is a downstream consequence of the intense oxidative stress in our model, and that its suppression by lycopene is potentially indirect, mediated through the enhancement of the cellular antioxidant defense via the Nrf-2 pathway. We also acknowledge the alternative possibility of a more direct modulation.

We believe this integration significantly strengthens the discussion by creating a more connected and plausible mechanistic framework for lycopene's protective effects.

4.The authors suggest that lycopene has clinical potential, which is reasonable, but the claim would be stronger with context. Discussing known human dosing, bioavailability, or prior clinical uses of lycopene would help bridge the gap between preclinical models and therapeutic application. As it stands, the claim lacks detail and feels speculative. A short paragraph bridging this translational gap would improve the discussion.

Response:

We sincerely thank the reviewer for this valuable suggestion to strengthen the clinical translation perspective of our discussion. We agree that providing context on human dosing and bioavailability is crucial for bridging the gap between our preclinical findings and potential therapeutic application.

As suggested, we have now added a new paragraph in the Discussion section that incorporates information on established human dosing, the challenge of lycopene's bioavailability, and potential formulation strategies (such as nanocarriers) that are being explored to overcome this limitation. This addition directly links our mechanistic results to the practical considerations for clinical development, thereby reducing the speculative nature of our original claim and providing a more grounded perspective on lycopene's therapeutic potential.

Minor Comments

1.Typos and terminologies

Response:

We sincerely thank the reviewer for their meticulous reading and for identifying these typographical and terminology errors. We have carefully corrected all the specific points raised throughout the manuscript:

1.a. "downregulation the PI3K/Akt axis" has been corrected to "downregulating the PI3K/Akt axis" in the Abstract (line 13).

1.b. "apoptpsis" has been corrected to "apoptosis" in the section title on page 21.

1.c. We have consistently used "H₂O₂" throughout the text and in all figures.

1.d. "lpy’s" has been corrected to "lycopene's" on page 25.

1.e. We have ensured proper spacing in all section headers, including "Immunohistochemical (IHC) Staining" on page 14.

1.f. "Elisa assays" has been corrected to "ELISA assays" on page 15.

We have also taken this opportunity to perform a thorough proofread of the entire manuscript to ensure consistency and accuracy in terminology and formatting.

2.Figure Interpretation and clarity:

Response:

We sincerely thank the reviewer for their thorough review and valuable comments regarding the clarity and interpretation of our figures. We have carefully addressed each point raised, as detailed below:

2.a We apologize for the ambiguity. In Figure 3A, the values for Bcl-2 and Bax indeed represent their individual protein expression levels, normalized to the internal control (e.g., β-actin), rather than their ratio. We have now clarified this explicitly in the revised figure legend and the corresponding Results section to avoid any confusion.

2.b The reviewer raises a valid point. We have double-checked the data and methodology for Figure 6C-I. The reported ~100-fold difference in ΔΨm (mitochondrial membrane potential) between control and treated groups is not an error but stems from the specific unit and normalization method we employed. The values were normalized to the control group set as a baseline. We have now added a detailed explanation in the Figure 6 legend and the Methods section regarding the normalization process and units used to ensure transparency and prevent misunderstanding.

2.c Thank you for highlighting this inconsistency. The images in Figure 7B are indeed from immunofluorescence (IF) staining performed on HK2 cells (a human kidney proximal tubular cell line) in vitro, not from tissue-based immunohistochemistry (IHC). We apologize for the incorrect terminology in the original label. We have now corrected the label for Figure 7B to "Immunofluorescence (IF) Staining in HK2 Cells" throughout the manuscript to accurately reflect the experimental method

3.Rephrasing Manuscript Title:

The title “Lycopene inhibits ER stress, apoptosis and increases AKT/PI3K and antioxidant-related proteins” is confusing as lycopene reduces phosphorylated (active) PI3K and AKT, it should be rephrased.

Suggestion: “Lycopene inhibits ER stress and apoptosis while modulating PI3K/AKT and enhancing antioxidant and anti-apoptotic proteins.”

Respons:

We thank the reviewer for this insightful comment and for suggesting a more accurate title. We agree that the original title could be misleading regarding the specific modulatory effect of lycopene on the PI3K/AKT pathway. We have therefore revised the title of the manuscript to adopt the reviewer's excellent suggestion,We believe this new title more precisely and clearly reflects our findings.

"Lycopene inhibits ER stress and apoptosis while modulating PI3K/AKT and enhancing antioxidant and anti-apoptotic proteins."

Reviewer 2 Comments to the Author

1.There are details missing in the entire manuscript. The introduction could have started with a generic problem and then discussed in detail.

Respons:

We thank the reviewer for the valuable suggestion to enhance the background and discussion in the Introduction. In response, we have now added a new opening paragraph that situates our study within the broader clinical context of sepsis and its impact on kidney function before delving into the specific molecular mechanisms. This addition begins with the general epidemiological and clinical problem of sepsis-associated acute kidney injury (SA-AKI), highlighting its prevalence, mortality burden, and current therapeutic limitations, thereby providing a clearer and more compelling rationale for our investigation. We believe this revision improves the logical flow from the general clinical problem to our specific scientific aims and experimental model, strengthening the overall narrative of the Introduction.

2.The details of mortality rates are missing. The authors say high mortality. Better to use stats from literature.

Respons:

We thank the reviewer for this valuable suggestion. In response to the comment regarding mortality rates, we have revised the opening statement in the Introduction by incorporating specific statistics from the literature to substantiate the claim of "high morbidity and mortality" in sepsis-associated acute kidney injury (SA-AKI). The revised sentence now reads:

"Sepsis-associated acute kidney injury (SA-AKI) is a lethal complication, as evidenced by a prospective study reporting a 28-day mortality rate of 32.7% in septic patients with AKI."

We believe this revision, supported by a concrete clinical statistic, strengthens the clinical relevance and impact of our study's background.

3.The authors have abruptly started using the abbreviated version for lycopene, withou

---

## [Decision Letter · Decision Letter 1]

9 Dec 2025

Lycopene inhibits ER stress and apoptosis while modulating PI3K/AKT and enhancing antioxidant and anti-apoptotic proteins

PONE-D-25-23053R1

Dear Dr. Sun,

We’re pleased to inform you that your manuscript has been judged scientifically suitable for publication and will be formally accepted for publication once it meets all outstanding technical requirements.

Kind regards,

Ajit Prakash, PhD

Academic Editor

PLOS ONE

Additional Editor Comments (optional):

Reviewers' comments:

Reviewer's Responses to Questions

**Comments to the Author**

Reviewer #1: All comments have been addressed

Reviewer #2: (No Response)

2. Is the manuscript technically sound, and do the data support the conclusions?

Reviewer #1: Yes

Reviewer #2: Yes

3. Has the statistical analysis been performed appropriately and rigorously?

Reviewer #1: Yes

Reviewer #2: Yes

4. Have the authors made all data underlying the findings in their manuscript fully available?

Reviewer #1: Yes

Reviewer #2: Yes

5. Is the manuscript presented in an intelligible fashion and written in standard English?

Reviewer #1: Yes

Reviewer #2: Yes

Reviewer #1: (No Response)

Reviewer #2: (No Response)

**Do you want your identity to be public for this peer review?** For information about this choice, including consent withdrawal, please see our Privacy Policy

Reviewer #1: No

Reviewer #2: No

---

## [Editor Report · Acceptance letter]

PONE-D-25-23053R1

PLOS One

Dear Dr. Sun,

I'm pleased to inform you that your manuscript has been deemed suitable for publication in PLOS One. Congratulations! Your manuscript is now being handed over to our production team.

Kind regards,

on behalf of

Dr. Ajit Prakash

Academic Editor

PLOS One